# Factors Affecting the Cases and Deaths of COVID-19 Victims

**DOI:** 10.3390/ijerph18020674

**Published:** 2021-01-14

**Authors:** Jerald M. Velasco, Wei-Chun Tseng, Chia-Lin Chang

**Affiliations:** 1Department of Applied Economics, National Chung Hsing University, Taichung 402, Taiwan; tweichun@nchu.edu.tw (W.-C.T.); changchialin@email.nchu.edu.tw (C.-L.C.); 2Department of Finance, National Chung Hsing University, Taichung 402, Taiwan

**Keywords:** COVID-19, populations, tests, temperature, cases, deaths

## Abstract

This paper attempts to find the factors that affect the number of cases and deaths of coronavirus disease 2019 (COVID-19) patients a year after the first outbreak in Wuhan, China. There were 141 countries affected with COVID-19 involved in the study. Countries were grouped based on population. Using ordinary least squares regression, it was found that the total number of cases and deaths were significantly related with the levels of population of the different countries. On the overall, median age of the country, and average temperature are positively related with the number of deaths from the virus. On the other hand, population density is positively related with the deaths due to COVID for low populated countries. The result of this preliminary study can be used as a benchmark for authorities in the formulation of policies with regards to treating COVID-19 related issues.

## 1. Introduction

In the midst of pandemic caused by severe acute respiratory syndrome–coronavirus 2 (SARS-CoV-2), now popularly referred to as coronavirus disease 2019 (COVID-19), many strategies have been adopted by governments as to how to fight this illness that has affected many lives. Non-pharmaceutical interventions (NPIs) are being implemented to eradicate the disease but these have just delayed and moderated the spread of the virus since vaccines and antiviral drugs have yet to be discovered [1]. A variety of regulations has been designed, among which include restriction of non-essential travel, imposition of social distancing, declaration of lockdowns on big cities and banning mass gatherings, all supposedly to prevent the spread of the novel corona virus [2].

The COVID-19 disease was first identified in Wuhan, China, in early December 2019 with the early cases reported in the city resulting in its spread worldwide that brought negatively significant changes to the normal lives of people in different countries of the world. Different laboratories and research institutions correspondingly responded and are in a race to discover solutions or a vaccine for the cure of this virus. As a result, there are more than 50 COVID-19 vaccine candidates under clinical trials [3].

As of 12 December 2020, there were already 71,437,016 cases recorded worldwide that resulted to 1,601,163 casualties [4]. There were 49,636,518 people who had already recovered from the disease not because of vaccines developed but only with the use of treatment for the symptoms of the patients. At this time of crisis, where vaccines were limited and most of them are in clinical trials, early prevention for such diseases is better than cure. Many states already conducted mass testing to determine the presence of viruses in different areas. But there are some arguments that mass testing is not necessary for their country. In South Korea, exploratory data analysis was used in assessing how effective mass testing is. According to the result, they noticed that 33 days after the first infection in the country, a significant increase in the number of tests resulted from a significant increase in the identified number of infected cases. This result allows the government to take early actions to focus more efforts in containing the breakout that will eventually happen [5]. According to Facundo, P. and Shi L., testing is a very close substitute for quarantine and can substantially reduce the need for indiscriminate quarantines [6]. With this, some rich countries have already administered mass testing like Germany, which carried out at least 500,000 tests a week as soon as the virus entered its territory [7]. However, some policy makers also says that infection rate does not justify mass testing there is no need for mass testing for COVID-19 in Taiwan as the rate of infection so far has been quite low [8]. Though in different countries have different scenarios, where number of cases and death rates varies, it is important to know what the different factors which may affect these numbers are.

Many researches have been published regarding the behavior, effects on humans, development of vaccines and etc., but not much research in assessing the number of deaths and cases of the said virus. This paper aims to provide an answer to the question: should a country provide mass testing? Is it necessary in our fight against COVID-19? Basically, this study investigates the factors affecting the number of deaths and cases of COVID-19 a year since the 1st outbreak occurred in Wuhan, the capital of the province of Hubei, China.

### Literature Review

A pandemic includes diseases of very different etiologies that exhibit a variety of epidemiologic features as applied to important global events spanning many centuries [9]. In the past few years, many diseases have appeared and threatened all the people worldwide. Before the appearance of this severe acute respiratory syndrome coronavirus 2 (SARS-CoV-2), some countries had already suffered from severe acute respiratory syndrome coronavirus (SARS-CoV) which is an infectious disease that caught global attention in the 21st century [10] and Middle East respiratory syndrome coronavirus (MERS-CoV). Asia and other parts of the world were affected by SARS-CoV in 2003 while MERS-CoV had a first outbreak in Saudi Arabia (2012 and 2018) and larger outbreaks occurred in South Korea in 2015 [11].

To avoid the wide spread of those diseases, equipment and new tools were developed to detect people who are showing symptoms either SARS or MERS. In 2003, there was no vaccine developed yet for SARS, but it was recommended by the World Health Organization (WHO) [12] that containment measures are based on: surveillance network including early warning system; isolation of suspected or probable cases, voluntary or mandatory quarantine of suspected contacts for 10 days and exit screening for outgoing passengers from areas with recent local transmission by asking questions and temperature measurement. Infrared (IR) cameras were also used to detect subjects with fever, the basic symptom of SARS and avian influenza. However, the accuracy of this system can be affected by human, environmental, and equipment variables. It might also be inconvenient that the thermal imager measures the skin temperature and not the core body temperature [13].

South Korea suffered from the MERS outbreak in 2015. According to Kim K.H., main contributors to the outbreak were late diagnosis, quarantine failure of ‘super spreaders’, familial care-giving and visiting non-disclosure by patients, and poor communication by the South Korean government, inadequate hospital infection management, and ‘doctor shopping’ [14]. The outbreak was entirely nosocomial, and was largely attributable to infection management and policy failures, rather than biomedical factors. Using multivariate regression analyses, risk factors such as black race is a risk factor for worse COVID-19 outcome independent of comorbidities, poverty, access to health care, and other mitigating factors. However, factors affecting case load include lower daily temperatures but do not affect deaths [15].

From the previous studies, many factors can be linked in the deaths and cases of COVID-19. In view of the environment, important factors affecting the mortality of the disease can be temperature variation and humidity [16]. However, there is no evidence supporting the assertion that case counts of COVID-19 could decline when the weather becomes warmer [17]. It was also found that there is a significant relationship between air pollution and COVID-19 infection after controlling for confounding factors [18]. In China, airflow direction of and air condition was consistent with droplets of the virus which cause more transmissions in restaurants [19]. In view of sociodemographic factors, number of cases, sex ratio, disposable income and the average age are significant determinants of the death rates in the German states [20]. COVID-19 deaths are mainly observed among elderly males aged over 65, and smoking patients might face a greater risk of developing into the critical or mortal condition and the comorbidities such as hypertension, diabetes, cardiovascular disease, obesity, population density and respiratory diseases could also greatly affect the prognosis of the COVID-19 [21,22,23,24,25,26,27,28]. In contrast, some researchers found that population density is not an important factor affecting COVID-19 cases or deaths [29,30].

Rate of infection and the fast transmission of the disease is one of the alarming issues of the pandemic. In Thailand, the number of tourists and their activities were found to be significant factors on the number of infected patients from COVID-19 [31]. In Italy, a tighter lockdown, mobility decreased enough to bring down transmission promptly below the level needed to sustain the epidemic [32]. However, new measures of social isolation are being implemented again as Italy experiences a second wave due to the easing preventive measures [33].

According to the Centers for Disease Control and Prevention (CDC) COVID-19 Response, geographic differences in reported case fatality ratios might also reflect differences in testing practices; jurisdictions with relatively high proportions of deaths might be those where testing has been more limited and restricted to the most severely ill [34].

As of today, determination of COVID-19 is now faster with the aid of a more robust quarantine strategy due to the lessons learned from SARS in 2003 [35]. However, scalable, rapid and affordable COVID-19 diagnostics could help to limit the spread of SARS-CoV-2, consequently saving lives [36].

## 2. Methodology

### 2.1. Gathering of Data

Information with regards to COVID-19 situations were collected from Worldometer, a website run by an international team of developers, researchers, and volunteers with the goal of making world statistics available in a thought-provoking and time relevant format to a wide audience around the world (https://www.worldometers.info/coronavirus). Information such as demographic characteristics of each country including its gross domestic product were obtained from the same data source. Weather information were collected from the website of WeatherBase then average temperature of the countries and its rainfall were included in the model (Table 1 and Table 2).

### 2.2. Ordinary Least Squares Estimates

In this study, ordinary least squares (OLS) regression were used to identify which factors might affect the number of cases and deaths from COVID-19. The model is assumed as follows:*Y_it_* = *α*_0_ + *β_i_X_i_* + *u_i_*,
where *i* denote the individual indexes of the country. *Y* is number of cases and number of deaths from COVID-19, and *X* is a set of independent variables, e.g., tests per million population (*T_PERM*), total tests (*TT*), tests per case (*T_PERC*), deaths per million population (*D_PERM*), population density (*PD*), median age of the country (*AGE*), rural population (*RUPOP*), GDP per capita (*GDP_PERC*), raw mortality rate (*RMR*), average temperature (*TEMP*) and average rainfall (*RAINF*). *α*_0_ is the intercept term and *β* is the slope of the estimated coefficients. The variable u, called the error term or disturbance in the relationship, represents factors other than *x* that affect *y*. (for more details, see Wooldridge 2013 [40]. Raw population density is a poor parameter and the best studies use weighted or lived population density e.g., [25].

The EVIEWS version 10.0 software was used to perform the estimations. To handle the heteroskedasticity and autocorrelation, the heteroskedasticity- and autocorrelation-consistent (HAC) robust covariance and standard errors were applied. Since there were two Y variables, the regression analysis was conducted separately. Empirical model used for the study are the following:*Total COVID Cases* = *α*_0_ + *β*_1_*T_PERM* + *β*_2_*TT* + *β*_3_*T_PERC* + *β*_4_*D_PERM* + *β*_5_*PD* + *β*_6_*AGE* + *β*_7_*RUPOP* + *β*_8_*GDP_PERC* + *β*_9_*RMR* + *β*_10_*TEMP* + *β*_11_*RAINF* + *µ_i_*(1)
*Total COVID Deaths* = *α*_0_ + *β*_1_*T_PERM* + *β*_2_*TT* + *β*_3_*T_PERC* + *β*_4_*PD* + *β*_5_*AGE* + *β*_6_*RUPOP* + *β*_7_*GDP_PERC* + *β*_8_*TEM* + *β*_9_*RAINF* + *µ_i_*(2)

Variables pertinent to the tests in detecting COVID-19 virus such as *TT*, *T_PERM*, which was computed as total tests given divided by one million as represented by the population of the country and *T_PERC* were considered to be influencing factors on the independent variables of the study since it has something to do on the fast detection and isolation of the infected people. *D_PERM* was also analyzed as this might indicate that as many people die from the virus, the possibility of being infected of the other people is increased. *PD*, *AGE* and *GDP_PERC* were also included since it was found that these variables significantly affects the transmission of the virus as far as the social distancing and level of activity of people and their age is a concern [41,42]. *RUPOP* was computed as 1 minus the percentage of the urban population of the country. The *RMR* of 1000 population were also considered and lastly, weather variables such as *TEMP* and *RAINF* were captured since previous studies proves that these might influence our dependent variables [43].

## 3. Discussion of Results

### 3.1. Factors Affecting Number of Confirmed Cases from Coronavirus Disease 2019 (COVID-19)

The OLS regression analysis with HAC standard errors and covariance were applied with Bartlett kernel, and Newey-West fixed bandwidth = 4. The result of the estimation is presented in Table 3. It was found that variables such as test per million population, total tests and test per case are significant at 1% while temperature was significant at 5%. Test per million has a coefficient −0.56 which means that in every unit in test per million increase, there is a reduction on the number of cases about 0.516. Total test has coefficient value of 0.06 which signifies that in every unit of test conducted, there is an increase of 0.6 case on the total case. Test per case has a coefficient of −3374.8 which signifies a negative relationship on the number of cases. We can infer that the more test being conducted in relation to every cases detected, there is a decrease on the number of cases by 3374. From all of the variables in relation to tests, it can be inferred that as we detect more cases, necessary actions can be done like to isolate people with virus and local transmission will not occur anymore. Meanwhile, temperature has a coefficient of 30,667.3 which indicates that the temperature causes more cases if this is being increased.

### 3.2. Factors Affecting Number of Deaths from COVID-19

Result from the second model can be gleaned in Table 4. At 1% level of significance, test per million, total tests, test per case are significant variables on the number of deaths from COVID-19. The coefficient of test per million was −0.016 which specifies that in every unit of test per million population death from the virus was reduced into 0.016. For the total tests, only 0.001 death occur if we increase the total test by one unit. When the tests/confirmed cases ratio increased with one unit, it leads to 68.44 deaths reduction. Meanwhile, Age, Rural population, and temperature are found to be significant at 10%. The coefficient value of the median age of the country is 365 which means that as people grow older in particular area number of death is increased by 365. Rural population can be interpreted such that in every percentage increase on this variable causes a reduction of 236 deaths. For the temperature, an additional 648 death case will be added if there is an increase in every unit of degree Celsius.

### 3.3. Cases and Deaths from COVID-19 in Countries with High and Low Population

Countries were group together according to the number of population to compare the variables which factors significantly affect their COVID-19 situations. Those countries with a population of 10 million and above were defined as the high population group while countries with population below 10 million were defined as the low population group. There were at least 77 countries who belongs to the high population group and a total of 64 countries for low population.

Comparing the two groups, it can be observed that in high population countries variables that are significant at 1% tests per million, total tests and tests per case with a coefficient of −2.434, 0.065, and −4275.8, respectively (Table 5). While low population country has a significant variables of Test per million (−0.22) and total test (0.03). Tests per case appeared to be non-significant variable. Based from the figures, there is less effect of tests per million on the number of cases in a low population countries. While a little bit higher effects of total tests in a high population area.

Meanwhile, population density and rural population were found to be significant factors at 5% on the number of cases for the low population areas. Population density has a coefficient of 75.6 which means that an increase in population density causes an additional 75.6 cases. Rural population on the other hand has a negative relationship on the number of cases, a total of 77,901.50 will be reduced if one percentage in rural population is added. Looking at the goodness of the fit of the model, R^2^ for the high population countries has a value of 0.87 which means that 87% of the total variation is explained by the factor included in the regression analysis. The value of R^2^ of low population groups was 0.66.

For the number of deaths, the results of the regression of the second model comparing the two groups of countries is presented at Table 6. In this estimation, at 1% level of significance variables that significantly affect the number of death from COVID-19 for the high population countries are total tests and tests per case with a coefficient of 0.01 and −82.55, respectively. On the low population countries’ tests per million, total tests, tests per case and age were found to be significant at 1%. As far as total test effects is concerned, two groups have the same negative sign on the coefficient and but are dissimilar in terms of coefficient. The effect of tests per case is lower in low populations with a value of −3.4 comparable to higher populations with a coefficient of −82.55. As the age of people increase it signifies that there are additional deaths of 72.94. At 10% level of significance, population density has an effect 0.95 increase in deaths as its unit increases and rural population have been found to be negatively significant with 12.94 death cases reduction if 1 more percentage people resides on the country side. 

The R^2^ value was observed to be lower at low population areas than higher population groups with a value of 0.46 and 0.71, respectively.

## 4. Conclusions

In view of the series of estimations conducted, the researchers draw the following conclusions: first, COVID-19 tests play an important role in our battle against this pandemic. The relationship of tests on cases and deaths were consistently significant even in countries with different levels of population. Carrying out more tests allows the authorities to detect more infections and do all the important steps in preventing infected people to have local transmissions. Although tests were also positively associated with deaths, its coefficient was good enough, which indicates that less deaths will occur as long as we do the isolation due to the early detection of the infected people. Second, age is a significant factor on the number of deaths. The findings shows that it is very important to protect older people from the exposure to more people as they are at risk of being infected by the virus. Third, rural population has negative effects on death. This is an advantage for countries which have a larger percentage of population in the countryside. This gives more people space and lessens the risk of catching COVID-19. Fourth, the temperature was positively significant for both number of cases and deaths. The mean temperature in our observation is 16.39 degree Celsius. Although +14.51 °C temperature is the favorable range for COVID-19 growth [44], temperature that is favorable for people to move out increases their chances to be exposed to the virus. A strict implementation of the safety protocols must be undertaken. The effect of population density was observed both on high and low population countries as significant factors on deaths. This is evidence that anywhere in the world, the higher the population density the riskier the threat of the virus.

Based on the results, it is suggested that the government may consider allocating a higher budget to testing equipment and facilities to fight this pandemic victoriously. A strict implementation of protocols with regards to social distancing and wearing of face masks is also advised in order to lessen the transmission of the virus. Nevertheless, each country may have its own best strategy depending on the availability of resources and funds that they have. The number of confirmed cases is globally underestimated, and comparisons across countries are difficult to determine due to differences in data collection procedures or health policies, among others. Thus, this statistical analysis of COVID-19 data needs to be interpreted with caution.

## Figures and Tables

**Table 1 ijerph-18-00674-t001:** Summary of data source.

Variables	Source
Total Case per country; Total Death per country; Total Test conducted	Worldometer [4]
Population Density; Median Age of the country; Urban Population	Worldometer [37]
Gross Domestic Product	Worldometer [38]
Average Temperature; Average Rainfall	WeatherBase [39]

**Table 2 ijerph-18-00674-t002:** Descriptive statistics of the variables.

Variables	Mean	Median	Maximum	Minimum
Total Case/Million	18,670.46	14,030.00	63,947.00	86.00
Tests/Million	22,0626.70	14,1031.00	1,423,225.00	2054.00
Total Tests	9,533,267.00	2,063,450.00	2.15 × 10^8^	50,488.00
Tests/Case	16.17	10.57	82.16	2.46
Deaths/Million	379.91	308.00	1516.00	3.00
Population Density	139.65	87.00	1380.00	3.00
Age	33.52	33.00	48.00	15.00
Rural Population	0.33	0.31	0.83	0.02
GDP/Capita	16,779.46	9881.00	80,296.00	376.00
Temperature	16.39	17.00	40.00	−0.6
Precipitation	874.18	773.10	2667.10	49.50

**Table 3 ijerph-18-00674-t003:** Result of ordinary least squares (OLS) regression analysis on factors affecting number of cases of coronavirus disease 2019 (COVID-19, based on 10 December 2020 data).

Variable	Coefficient	Std. Error	t-Statistic	Prob.
Constant	−670,837.4	405074.2	−1.656	0.100
Tests/Million	−0.57	0.21	−2.661	0.009 ***
Total Tests	0.06	0.007	8.471	<0.001 ***
Tests/Case	−3374.81	1300.57	−2.595	0.011 ***
Deaths/Million	22,659.64	158,355.8	0.143	0.886 ^ns^
Population Density	8.39	97.98	0.086	0.932 ^ns^
Age	5801.06	6527.42	0.889	0.376 ^ns^
Rural Population	−240,669.9	39,8913	−0.603	0.547 ^ns^
GDP/Capita	8.65	5.35	1.617	0.108 ^ns^
Raw Mortality Rate	−22,056,641	158,000,000	−0.139	0.890 ^ns^
Temperature	30,667.33	13,586.81	2.257	0.026 **
Rainfall	76.80	82.28	0.933	0.352 ^ns^
R-squared	0.829	F-statistic	56.806	
Adjusted R-squared	0.814	Prob(F-statistic)	<0.001	

Note: ***-Significant at 1%; **-Significant at 5%; ^ns^–not significant.

**Table 4 ijerph-18-00674-t004:** Result of OLS regression analysis on factors affecting number of deaths of COVID-19 (10 December 2020 data).

Variable	Coefficient	Std. Error	t-Statistic	Prob.
Constant	−4989.98	9446.07	−0.528	0.598
Test/Million	−0.016	0.007	−2.446	0.016 ***
Total Test	0.001	0.00	7.105	<0.001 ***
Test/Case	−68.44	22.62	−3.025	0.003 ***
Population Density	−3.10	3.38	−0.918	0.361 ^ns^
Age	365.04	208.84	1.748	0.083 *
Rural Population	−23,696.37	13,939.86	−1.700	0.092 *
GDP/Capita	0.10	0.11	0.862	0.391 ^ns^
Temperature	648.08	356.77	1.817	0.072 *
Rainfall	0.17	1.85	0.092	0.927 ^ns^
R-squared	0.690	F-statistic	32.370	
Adjusted R-squared	0.669	Prob(F-statistic)	<0.001	

Note: ***-Significant at 1%; *-Significant at 10%; ^ns^–not significant.

**Table 5 ijerph-18-00674-t005:** Result of OLS regression analysis on factors affecting number of cases of COVID-19.

	High Population Areas				Low Population Areas			
Variable	Coeff.	S. E.	t-Stat.	Prob.	Coeff.	S. E.	t-Stat.	Prob.
Constant	−1,527,543	786,425.2	−1.942	0.056 ^ns^	97,110.06	48,850.86	1.988	0.052
Test /Million	−2.434	1.38	−1.760	0.083 **	−0.22	0.07	−3.048	0.004 ***
Total Test	0.065	0.007	9.779	0.00 ***	0.03	0.008	3.781	<0.001 ***
Test/Case	−4275.8	1515.14	−2.822	0.006 ***	−87.79	55.52	−1.581	0.120 ^ns^
Death/Million	65,324.64	29,6146.5	0.221	0.826 ^ns^	4171.76	23,887.39	0.175	0.862 ^ns^
Pop. Density	−257.49	333.54	−0.772	0.443 ^ns^	75.60	28.84	2.622	0.011 **
Age	16,555.06	11,249.27	1.472	0.146 ^ns^	108.29	1231	0.088	0.930 ^ns^
Rural Population	202,975.9	577,967.10	0.351	0.727 ^ns^	−77,901.50	37,918.24	−2.054	0.045 **
GDP/Capita	17.33	10.84	1.599	0.115 ^ns^	1.22	0.73	1.665	0.102 ^ns^
R M R	−64,046,492	296,000,000	−0.216	0.830 ^ns^	−4,024,853	2,390,6984	−0.168	0.867 ^ns^
Temperature	52,773.07	25,077.25	2.104	0.039 **	−1532.85	1365.54	−1.123	0.267 ^ns^
Rainfall	45.37	120.20	0.377	0.707 ^ns^	−11.95	8.98	−1.331	0.189 ^ns^
R^2^	0.874	F-stat.	41.157	R^2^	0.668	F-statistic		9.540
Adj. R^2^	0.853	Prob.	<0.001	Adj. R^2^	0.598	Prob(F-statistic)		<0.001

Note: ***-Significant at 1%; **-Significant at 5%; ^ns^–not significant.

**Table 6 ijerph-18-00674-t006:** Result of OLS regression analysis on factors affecting number of deaths of COVID-19.

	High Population Areas				Low Population Areas			
Variable	Coeff.	S.E.	t-stat.	Prob.	Coeff.	S. E.	t-stat.	Prob.
Constant	−5375.87	15,519.28	−0.346	0.730	180.55	898.98	0.201	0.842
Tests/M	−0.02	0.02	−0.748	0.457 ^ns^	−0.004	0.001	−3.240	0.002 ***
Total Tests	0.001	0.00	8.519	<0.001 ***	0.0004	0.00	4.047	<0.001 ***
Tests/Case	−82.55	24.56	−3.361	<0.001 ***	−3.45	0.91	−3.809	<0.001 ***
Pop. Density	−13.69	10.66	−1.284	0.204 ^ns^	0.95	0.51	1.877	0.066 *
Age	373.21	382.80	0.975	0.333 ^ns^	72.95	24.35	2.996	0.004 ***
Rural Pop.	−35,365.37	22,396.51	−1.579	0.119 ^ns^	−1294.06	733.14	−1.765	0.083 *
GDP/Capita	0.20	0.28	0.718	0.475 ^ns^	0.008	0.01	0.548	0.586 ^ns^
Temperature	1129.91	683.76	1.653	0.103 ^ns^	−42.89	26.06	−1.646	0.106 ^ns^
Rainfall	0.21	4.20	0.052	0.959 ^ns^	0.17	0.16	1.081	0.285 ^ns^
R^2^	0.721	F-stat.		19.257	R^2^	0.468	F-stat.	5.280
Adj. R^2^	0.684	Prob.(F-stat)		<0.001	Adj. R^2^	0.379	Prob (F-stat)	<0.001

Note: ***-Significant at 1%; *-Significant at 10%; ^ns^–not significant.

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
