# Peer review of "Factors Affecting the Cases and Deaths of COVID-19 Victims"

_ijerph, 2021, doi:10.3390/ijerph18020674_

Round 1
Reviewer 1 Report
The manuscript in question is a new updated version of another manuscript that I have previously revised.
The changes were substantial. The suggestions made previously were accepted and carried out, as well as the corrections noted in the previous manuscript.
The manuscript has evolved positively in its structure.
Therefore, I recommend the manuscript for acceptance and subsequent publication.
Reviewer 2 Report
A much improved paper.
Line 50, 50,000 or 500,000 ????
Line 101, obesity is also a major risk actor
Literature review, no mention of population density although I know of at least 4 references
Line 153, need to make the comment that raw population density is a poor parameter and the best studies use weighted or lived population density
Line 245, Table not table
Line 261, add space
Author Response
Please see the attachment.

This manuscript is a resubmission of an earlier submission. The following is a list of the peer review reports and author responses from that submission.
Round 1
Reviewer 1 Report
My considerations are in the attached file.

Author Response
Dear IJERPH Editorial Office,
Point 1: Please note that references are cited in an incorrect pattern so please revise them before submission of the manuscript. For example [1]
Response: Attached is the revised manuscript following your comments so it can be processed further. In the revision, you could notice that citations were numbered using [ ]. From 24 reference, it was turned 25. Number 25 was not included in the original version. Trying my best to satisfy the requirements, references was also revised based on the guidelines you have sent to me.
Thank you and more power.
Regards
Jerald

Reviewer 2 Report
Velasco and Tseng used Ordinary Least square regression to analyze factors affect COVID-19 cases number and death toll. Case number was affected significantly by total cases per million, new cases, total test, active cases, death per million, new death, and total test per million. Total death numbers are affected by active cases, death per million, total cases per million, and new death. The analysis model is well-designed and the results support the conclusion. Similar approaches can be applied to analyze recent COVID-19 case surge.
Comments:
- The manuscript needs to be edited and proofread by native English speaker with biological background.
- Change Covid-19 to COVID-19 throughout the manuscript.
- Line 27: add the website for real time COVID-19 update https://www.worldometers.info/coronavirus/
- Does Johns Hopkins COVID-19 data (https://coronavirus.jhu.edu/map.html) include more countries worldwide? More than 68?
Reviewer 3 Report
The authors have rushed into the application of correlation without thought.
Explanatory variables for cases should be population age structure,
perhaps try raw mortality rate (deaths/1,000 population) available from
the World Bank and other sources.
Population density has been shown to play a role in the spread of the
virus (several papers in IJERPH). Maybe try percentage rural population which I think is available from the world bank.
Other groups have shown that the number of Covid tests performed are
influenced by GDP. GDP per head or similar should be tried as an
explanatory variable
Detailed data is available for US states from the Covid tracking
project The COVID Tracking Project | The COVID Tracking Project
https://covidtracking.com/
Does the model work with the US data?
Mortality rates from Covid-19 are known to be a log function of age.
Try grouping similar countries together or grouping with similar case fatality rates. Are there any similarities.
Case fatality rates have reduced over time. Try comparing data to the
end of June with more recent data.
The authors need to think through all possible explanatory variables
before blindly applying regression.
>
> Average levels of pollution must be available, average temperatures are
> available, hours of sunshine, average humidity should all be available.
>
> At the moment the paper is below average but has the potential to be
> significantly bettwe,